# Living Lab Experience in Turin: Lifestyles and Exposure to Black Carbon

**DOI:** 10.3390/ijerph19073866

**Published:** 2022-03-24

**Authors:** Ornella Salimbene, Luca Boniardi, Andrea Maria Lingua, Marco Ravina, Mariachiara Zanetti, Deborah Panepinto

**Affiliations:** 1DIATI—Department of Environment, Land and Infrastructure Engineering, Politecnico of Torino, 10129 Torino, Italy; andrea.lingua@polito.it (A.M.L.); marco.ravina@polito.it (M.R.); mariachiara.zanetti@polito.it (M.Z.); deborah.panepinto@polito.it (D.P.); 2EPIGET—Epidemiology, Epigenetics, and Toxicology Lab, Department of Clinical Sciences and Community Health, University of Milan, 20122 Milan, Italy; luca.boniardi@unimi.it

**Keywords:** living laboratory, air pollution, black carbon, personal exposure, time-activity pattern, microenvironments

## Abstract

State-of-the-art, continuous personal monitoring is a reference point for assessing exposure to air pollution. European air-quality standards for particulate matter (PM) use mass concentration of PM (PM with aerodynamic diameters ≤ 10 μm (PM10) or ≤2.5 μm (PM2.5)) as the metric. It would be desirable to determine whether black carbon (BC) can be used as a better, newer indicator than PM10 and PM2.5. This article discusses the preliminary results of one of the three living laboratories developed in the project “Combination of traditional air quality indicators with an additional traffic proxy: Black Carbon (BC)”. The Living Lab#1 (LL#1) involved 15 users in the city of Turin, Italy. Three portable aethalometers (AE51) were used to detect personal equivalent black carbon (eBC) concentrations in the respiratory area of volunteers at 10-s intervals as they went about their normal daily activities. The Geo-Tracker App and a longitudinal temporal activity diary were used to track users’ movements. The sampling campaign was performed in November for one week. and each user was investigated for 24 h. A total of 8640 eBC measurements were obtained with an average daily personal exposure of 3.1 µg/m^3^ (±SD 1.3). The change in movement patterns and the variability of microenvironments were decisive determinants of exposure. Preliminary results highlight the potential utility of Living Labs to promote innovative approaches to design an urban-scale air-quality management plan which also includes BC as a new indicator.

## 1. Introduction

The air quality in many parts of Europe, and chiefly in the zones where the conditions for dispersion of pollutants are less favorable (e.g., Northern Italy, The Netherlands), presents critical conditions for the health of the exposed populations [1]. Although several natural sources (volcanoes, fire, etc.) may release pollutants in the environment, anthropogenic activities are the major cause of environmental air pollution and represent a continuous hazard to human health. The WHO has labelled air pollution as the major environmental threat to health and has estimated that approximately 90% of the world population does not breathe air complying with Air Quality Guideline [2]. Administration, businesses, and the population need efficient indicators of the possible effects of a change in decision, strategy, or behaviors.

The monetary quantification of the health effects [3] of atmospheric pollution (through the definition of external costs) and carbonaceous components of ultrafine particles are increasingly recognized as useful indicators to support decisions and information at all levels. In a novel context, in which climate change and air quality become two related aspects of the same problem [4], it is necessary to carry out novel local strategies for Air Quality Monitoring Networks’ (AQMNs) innovation, not only applying analytical optimization procedures of the existing networks, but also introducing advanced devices by using data by remote investigations [5] and new indicators.

BC is recognized as a valuable indicator for the evaluation of local traffic control policies thanks to its strong spatial contrast inside the urban environment [6,7,8]. Moreover, it is recognized as an indicator of the health effects of air particulate matter in addition to PM2.5 [9] and as a climate-forcing agent [10,11], second only to carbon dioxide. For these reasons, BC should be more investigated inside urban areas by using living laboratories (LLs). The active involvement of researchers in complex real-life contexts is not yet well-developed in the realities of Italian metropolises. LLs attempt to address this problem.

Living environments with human protagonists present major challenges for re-searchers taking on the role of impartial intermediaries to initiate and facilitate negotiations between stakeholders. Therefore, in this context, researchers represent Living Lab facilitators in multi-stakeholder environments [12]. Specialized fields of knowledge distinguish study and action activities in living, social, or organizational contexts (different environments, multi-locality and multi-stakeholder, groups of researchers from different backgrounds (technical or organizational)).

In recent years, the notion of a living lab has been constantly expanded and several European projects have been implemented which have emphasized the active role of users as co-innovators in air pollution-monitoring activities. In 2016, more than 2000 people living in Antwerp took part in the “Curieuze Neuzen” project [13], which aimed at mapping nitrogen dioxide street by street. The collected data was then used by the institutions to improve the accuracy of the Flemish official model of air quality (EEA, 2021).

In the same year, the AirCasting Brussels project represented an interesting example of a living lab conducted by both the academic world and civil society [14]. As part of the project, a group of 7 adults were actively involved in designing and running a 1-week personal PM2.5 monitoring campaign aimed at studying perceptions, beliefs, and exposure according to individual space–time trajectories [15]. At the national level, the research project “Mapping of air pollution in a school catchment area of Milan” (MAPS MI) involved parents, teachers, and children of a primary school in the city, as well as residents of the school catchment area of a research process aimed at mapping the airborne contaminant black carbon (BC) on a neighborhood scale [16], studying the determinants of personal exposure of school-age children [17], and identifying possible exposure hotspots during home–school journeys [18]. In 2018, the research project “Disordered sleep breathing and the relationship with PM10” involved 350 citizens suffering from obstructive sleep apnea syndrome (OSAS) and residing in Rome, Italy with the aim of investigating the correlations between severity of the OSAS and the concentration of PM10 in the different districts of the city [19].

In this context, it is important to distinguish the users directly involved (subjects involved in data collection), the beneficiaries and the stakeholders in the context of an intersectoral living lab in which technology suppliers, professionals, public and private companies, users and researchers are integrated for open and constructive innovation.

Users (local, regional communities, etc.) become the “key elements” for the implementation of new monitoring and data-acquisition strategies and for providing new research questions in the field of atmospheric monitoring [20].

This approach is particularly necessary in local situations wherein the problem of air pollutants is difficult to manage. Therefore, it is restrictive to use the data collected by national monitoring stations to study personal exposure to the main pollutants and design regulatory plans for the traffic. The Living Labs are configured as a new methodological approach that adds distinctive traits to the tradition of action research. The Living Lab Research Approach (LLRA) takes an evolutionary view of innovation and studies new technologies in complex real-world contexts [21]. The following article provides a summary of the LLAs used in the project “Combination of traditional air quality indicators with an additional traffic proxy indicator: Black Carbon” set in the metropolitan area of Turin in Northern Italy. Based on this topic, the project formulates three main research questions (Q):
(Q1) What possible data, tools, and methods could be strategic for an alternative co-design of urban air-quality management plans?(Q2) How can they be developed and adopted in local air-quality management policies to ensure the implementation of safety protocols in favor of citizens’ health?(Q3) How could citizen science be used for environmental monitoring and communication, and what are the potential benefits in terms of data interoperability?


To answer the previous research questions through an empirical approach, the project evaluates and models the standard air-quality indicators (PM10–PM2.5) with a new possible indicator: black carbon (BC) and proposes its implementation in air-quality management plans in terms of risk to citizens’ health. Figure 1 shows how the implementation of LLs’ information can increase knowledge on the real exposure of citizens in the city of Turin and can provide useful data to concretely answer research questions Q1, Q2, and Q3.

As shown in Figure 1, the individual LLs have been designed in such a way as to form a set of overlapping and partly sequential cases of continuous learning and knowledge extension, pursuing the concept of “Healthy Cities” (WHO, 2020) and strongly promoting and emphasizing participatory governance, solidarity, and intersectoral collaboration. LL#1 involved 15 volunteer citizens residing in Turin who were observed and monitored during a week (from 7 to 12 November 2021). The objective of the LL#1 pilot study is to collect information on the exposure of citizens on a neighborhood scale to superimpose the information in real time with that collected by the regional monitoring network. These data will allow the scientific community and local politicians to plan and co-design air-quality management plans on an urban scale, also integrating new indicators such as BC. Behavior, lifestyle, occupation, and daily personal exposure to BC during the 24 h were analyzed. LL#2 monitored for one year (October 2020–October 2021) the respiratory parameters of 50 male volunteers suffering from mild chronic obstructive pulmonary disease (COPD) and residing in areas with different traffic densities.

More precisely, 25 volunteers resided in highly trafficked areas and 25 resided in areas with low traffic density. LL#2 allowed us to investigate the associations between the concentrations of PM, BC, and Forced Expiratory Volume in the 1st second (FEV1). During this LL, data collected by the urban background control unit managed by ARPA and located in V. Monti (Lingotto, Turin) were used.

LL#2 showed that high traffic density is significantly associated with lower forced expiratory volume in 1 s (FEV1) and forced vital capacity (FVC). The results from LL#2 highlight that people with persistent respiratory symptoms and residing in busier areas appear to be more vulnerable to BC exposure and present greater nocturnal dyspnea. Finally, LL#3 involved a social survey on over 300 Turin citizens who participated in a semi-structured questionnaire on urban air quality and their behaviors adopted in favor of urban air quality (winter 2021). Each seemingly different LL is interconnected by a common denominator: data collected by users, such as active human sensors, provide information to be integrated into the management of air-quality regulatory policies at the local level (Figure 1). Through the multi-LL, it is possible to investigate the pollution hotspots in the different urban districts, the hours of the day when people are most exposed and during which activities, the behaviors that make citizens more vulnerable, their perception of air quality, how traffic interacts with the parameters of some respiratory diseases in patients treated at home, and the behaviors adopted by citizens in favor of air quality. This article describes in more detail the activities developed in Living Lab#1. 

It is essential to remember that pollutants such as BC are not regulated and monitored at the urban level in the city of Turin (considered in this case study), and there is only one BC monitoring unit, active since 2018 and managed by the Regional Environmental Protection Agency (ARPA).

In state-of-the-art research, there is great interest in determining whether BC functions as an indicator of the toxicity of other co-pollutants. There is evidence that black carbon is associated with asthma and other respiratory problems [22], low birth rates, heart attacks, and lung cancer [23].

On an urban scale, BC is a good tracer for traffic emissions (US-EPA, 2011) but is poorly measured, and the sensors for analysis are too expensive to be implemented in the general national monitoring network. This would seem inconsistent with the fact that the part of air-quality management policies in European cities is aimed at road-traffic mitigation strategies. A critical question in many urban environments, is not whether the air in cities is unhealthy, but rather to understand how severely affected the health of the population is. Answering this question is very difficult because the analysis of air-quality data is mainly linked to the specificity of the area studied and the appropriate indices for the evaluation of the air-quality component and the urban climate, have yet to be developed and rigorously tested in their entirety. Internationally, there is growing concern about the potential negative impact on the health of those living, attending school, or working in places near busy roads. This research project, through the activities of urban laboratory and participatory science (LL) captures the variability of BC exposure in real-life conditions. Exposure metrics thus become a useful tool for the scientific community when combined with environmental data, air-quality models, receptor models, and exposure models.

## 2. Materials and Methods: Urban Living Lab Research Design LL#1

Living Lab 1 was established in the city of Turin (Figure 2) and involved 15 users, including 11 people afferent to the Disaster Recovery Team (DIRECT Team) of the Geomatics Laboratory (Politecnico di Torino, Torino, Italy) and four external users. The following directives were suggested to the users: (a) fill in a diary of daily activities, (b) wear a shoulder strap containing an AE51 aethalometer for black carbon measurement, and (c) activate the Geo-Tracker app for the duration of the sampling. All logistical operations and the start of black carbon monitoring were carried out within the headquarters of the Turin Polytechnic in a dedicated classroom. People were divided into 5 groups and on each day of the week, a group of 3 users started monitoring between 19:00 and 20:00 wearing the sensor. The monitoring (for each tester) ended the next day between 19:00 and 20:00. Before participating in the activities, a briefing was organized to train volunteers on how to handle the equipment and how to behave. During indoor sports, users left the sensor close to their body. None of the users played outdoor sports.

### 2.1. Instruments and LL#1 Tools

During the exploratory experience of Living Lab#1, three AE51 micro-aethalometers supplied by the Department of Clinical and Community Sciences (DISCCO) of the State University of Milan, were used. The microAeth^®^ AE51 is a real-time pocket BC aerosol monitor that incorporates the same Aethalometer^®^ measurement technology that has been used around the world for over 30 years and is still used by the official Air Quality Network (AQN) worldwide. The air sample is collected on T60 filter media (Teflon-coated glass fiber), and concentrations are estimated by measuring the rate of attenuation (ATN) of an 880-nm wavelength beam of light passing through the same spot. BC levels estimated by the micro-aethalometer are usually referred to as equivalent black carbon (eBC) because they are derived by applying fixed factors such as an assumed mass absorption cross-section [24]. These tools have been shown to work both in terms of precision and accuracy, even compared to the tools of international institutional air-quality monitoring networks that use the same underlying technology [25,26].

In the context of the LL#1 pilot study, the devices were placed in a dedicated bag (Figure 3) and worn by the users in order to have the sampler tube as close as possible to the respiratory zone. All users charged the instrument overnight and left it in their room, next to the bed. In this way, the instrument detected concentrations in an area close to the person during the night.

The sample rate was set to 10 s and a flow of 150 mL/min. Users received the daily diary to be filled in according to a longitudinal chronological order, thus reporting a description of the number of trips made in 24 h, means of transport used, places frequented, domestic environments, and their respective characteristics (heating, presence of pets, exposure of houses on busy streets, type of kitchen used). Users have activated a GPS tracker system with a simple interface and with data export functionality (gpx file). To use the tracking system, each user has downloaded the Geo-Tracker app on their smartphone and activated the GPS for 24 h.

### 2.2. BC Data Handling

Raw data were post-processed as follows.

(1) The optimized noise-reduction algorithm (ONA) was applied to smooth background noise [27].

(2) A set of three fixed factors, one for each device, were applied to account for differences between the devices. These factors represented the regression slopes between BC concentrations measured by a gold standard (a MA200 micro-aethalometer recently calibrated by the manufacturer) and those from the AE51s as part of an intercomparison exercise conducted one week before the Living Lab#1 monitoring campaign.

(3) The so-called Virkkula algorithm with a k-correction factor equal to 0.007 was applied to correct the loading effect of the particulate matter collected on the filter [28,29].

### 2.3. Living Lab#1 Setting and Dataset

LL#1 was set in Turin, northern Italy (population 886.837). The main sources of BC contributing to personal exposure in this city are determined by domestic heating and vehicular traffic. During the LL#1, the average temperature in Turin was 8.4 °C, with light winds and one day of rainfall. Personal information (gender, age, occupation, smoking habit), data about the home (type of heating, exposure of the house, floor plan, type of street facing the house (road classification is based on open-street map), presence of pets, use of candles, distance from AQN station), number of daily trips and the means of transport used by the volunteers (bike, car, subway, foot, e-scooter, tram) are shown in Table 1. Table 1 also summarizes the average concentration of eBC, measured by the aethalometers during the night and during commuting activities These concentrations are compared with the average concentration measured by the urban background control unit managed by the ARPA.

LL#1 involved 11 females (f) and 4 males (m) with an average age of 30.2, all volunteers are employed except one student (ID04). ID13 is a smoker but did not use cigarettes during the monitoring activities. All the other volunteers are non-smokers, and three of them stated that they had stopped smoking for at least six months. Volunteers ID04 and ID14 live with a pet. No volunteers suffer from allergies or respiratory diseases. As for the heating system, 8 volunteers use an internal boiler (IB) installed in the kitchen and bathroom, and the others have an external boiler (EB). Some houses are equipped with district heating. Volunteers made most of their daily journeys on foot, car, or by bicycle, or the subway. The e-scooter and the tram were the least used means of transport.

The information reported in the diaries were manually compared with the data recorded by the Geo-Tracker carried out by each user on their smartphone. In this way, the paths of all users were recorded, and the eBC concentrations were superimposed on the gpx files to reconstruct the spatial trend of the concentrations. Most of the users–volunteers live near the Politecnico di Torino and attend university offices and laboratories located in the headquarters in Corso Duca degli Abruzzi and at the Valentino Castle. Each piece of eBC data (time resolution 10 s and aggregates per minute) was associated with the different environments frequented, the paths taken, and the activities carried out. The following macro-living environments (MEs) have been identified as linked to micro-environments and/or activities (µEs): (a) home, that is, the time spent at home working, cooking, relaxing, or sleeping; (b) indoor (other), that is, all frequented indoor environments other than home (gyms, bars, restaurants, workplace etc.); (c) transport, that is, all environments linked to circulation throughout the day, distinguishing between the means of transport used; (d) outdoor (other), that is, all those external environments, other than those related to transport, frequented during the day (parks, etc.). Statistical analysis was performed with RStudio. The non-parametrical Kruskal–Wallis test was used to check differences among groups. To control for the false discovery rate in the presence of more than two groups, a Benjamini and Hochberg post-hoc test was used [30].

Differences were identified as significant with *p* < 0.05. ANOVA was used to check statistical differences between days, and the intra-class correlation coefficient (ICC) was computed to identify the proportion of the total variability explained by both between- and within-days variability.

## 3. Results

Of the 24-h measurements triggered by users, 99% of the data was captured successfully. Users showed interest in monitoring activities and great participation. The measurements were made in 12 residential houses inhabited by non-smokers and in 3 houses with smokers (no house with a closed wood stove or fireplace, 8 houses with an internal boiler located in the kitchen or bathroom; the remaining houses were equipped with an external boiler). In one house (ID15, Figure 4e), householders used scented candles. Measurements were also carried out during transportation periods, inside cars (used by six users) mostly with fully closed windows and recirculation, as well as during walking, cycling, on the subway, on electric scooters, and on the tram.

The personal concentration time trends measured during the monitoring campaign are reported for each volunteer in Figure 4 along with specifications on activities and environments. Statistics of personal eBC are reported in Table 1 according to IDs as well as in Table 2 (with a focus on macro- and micro-environments/activities). In Table 2, eBC concentrations are given as mean, standard deviation (SD), and percentiles.

In Figure 4, the arithmetic mean of the 24-h exposure concentration of each ID is reported by the horizontal benchmark in black. The horizontal benchmark in red shows the arithmetic mean of the eBC concentrations measured for the same period at the Air Quality Network (AQN) urban background monitoring site (the only fixed BC measuring station in the city). Figure 4 shows the graphs related to the concentrations of eBC measured by the volunteers. Each box (a, b, c, d, e) represents the measurements of three volunteers (ID). The relative differences between AQN concentrations and measured personal concentrations ranged from −91% to +45%, whereas differences were between −10% and +10% in only 5 cases. Among the 15 volunteers, 5 (ID4, Figure 4b; ID7, Figure 4c; ID10, Figure 4d; ID13, ID15, Figure 4e) detected a daily average concentration distinctly above the AQN values. These values can be liked with the combined action of high pollution levels at home due to the presence of eBC sources (burning candles, tenants who smoke) and longer time spent in transportation (ID15, ID13, Figure 4e) or the use of polluted means of transport (ID4, (Figure 4b), or possibly with exposure events (ID7, Figure 4c at the restaurant). No significative differences (ANOVA *p* = 0.11) were found between days if considering exposures, and two-thirds of the total variability was explained by the within-day variance (ICC = 0.34). The distributions of eBC detected in each macro-living environment were significantly different from each other, with transport showing the highest values (mean ± sd equal to 4.8 ± 4.7), followed by other outdoor environments (4.0 ± 2.9), home (3.0 ± 2.2), and other indoor environments (2.6 ± 4.0).

Residential indoor nighttime mean concentrations were higher for ID15 (8.1 μg/m^3^, Figure 4e), ID10 (5.2 μg/m^3^, Figure 4d), ID15 (4.8 μg/m^3^), and ID2 (3.8 μg/m^3^, Figure 4a). The very high concentration measured by ID15 can be attributed to the use of burning scented candles. For ID13 (Figure 4e) and ID2, possible sources of pollution are represented respectively by smoke and restaurants’ chimneys located close to apartment windows. Finally, ID10 lived in a house at the ground floor partially facing a residential street, and this can be identified as the only source of pollution given the other information collected.

As reported in Table 2, the highest peaks of exposure were measured at the restaurants, and during transportation (especially while walking, commuting by car, and on the tram). High eBC concentrations were also recorded while cooking (especially during the preparation of roasts), and while spending time outdoor for general activities (waiting or chatting), probably due to the proximity to the traffic.

Users spent more than 90% of their time indoors, in residential and work (office) environments. ID7 (Figure 4c) experimented with smart-working in two different conditions: open windows and closed windows. Working with the windows closed in a seventh-floor house, he was exposed to eBC concentrations below 2 μg/m^3^, while working with the side-street windows open, the concentrations were found to be above 3 μg/m^3^ with peaks of 10.5 μg/m^3^, highlighting the role of fixture in preventing exposure episodes.

On average, volunteers spent 7% of time in transport. The highest mean eBC concentrations were detected on the tram, followed by the subway, commuting by car, walking, and biking. As previously reported, the highest peaks of exposure were detected while walking, commuting by car, and on the tram. By comparing the most representative means of transport (i.e., walking, commuting by car, and biking), exposure during walking was significantly different than during commuting by car and biking, while no significant differences were found between commuting by car and biking. Exposure concentrations measured during morning rush hour (6–9 AM mean ± SD equal to 5.8 ± 4.8 μg/m^3^) were significantly higher than those at midday (9 AM–5 PM; 4.8 ± 5.6 μg/m^3^), evening rush hour (5–9 PM; 4.5 ± 4.0 μg/m^3^), and night (9 PM–6 AM; 4.1 ± 2.7 μg/m^3^).

## 4. Discussion

Through LL#1, it was possible to measure the daily personal exposure of eBC of 15 users based on 8640 measurements in 24 h. The average daily exposure concentration was between 1.6 and 6.7 μg/m^3^. Overall, commuting represented a disproportionate share of daily exposure and reduced the contribution of time spent indoors and outdoors. In fact, there are marked effects of the mode of transport on the contribution of commuting to daily exposure. Mean 24-h exposure concentrations and measured microenvironmental levels are typically toward the lower end of the range reported in comparable studies [31,32,33,34]. The marked variability in eBC concentrations between the different modes of transport (bicycle, foot, tram, car) and the different microenvironments observed (offices, bars, restaurants, gyms, home) is consistent with other authors [33,34].

In this study, users’ trips were calculated, analyzing the modes of movement and the microenvironments of stay. The main commuting modality used by the different IDs (cycling and walking) seems to have an appreciable effect both on their daily exposure to eBC and on the contribution that commuting makes to it. The lowest exposure mode is likely to vary from place to place. Surely changing travel modes can be used to reduce total exposure if high- and low-exposure modes are identified (users who have cycled and walked recorded the highest exposure levels). Being in the office or at home led to nearly halving daily eBC exposure. Very crowded environments, such as a university classroom (ID4), have recorded peaks in eBC concentrations (16 μg/m^3^), so it could be useful to adopt more frequent recirculation and air-change systems in these crowded environments. In a specific case (ID7), the air exchange in the indoor environment had negative effects on the exposure in a house located on the seventh floor. Therefore, it is important to evaluate the exposure of the house, the presence of highly trafficked roads, the presence of emission sources close to the house (industries and chimneys). The results highlight how citizens’ behaviors can contribute to increasing or reducing daily exposure to eBC. Regarding home heating, it has been noted that in some cases older, low-maintenance boilers located in kitchens cause higher emissions. Additionally, people living in homes with IB during the heating season and living on the ground floor seem more exposed. LL#1 is a pilot survey and has important limitations. The number of measurements in different microenvironments was not equivalent. Furthermore, all measurements were collected from 15 users employed in offices and in privileged working conditions (office workers), which means that the results can only be partially representative of the exposures incurred by a larger population. The results presented here should be interpreted with these limitations in mind. It is important to underline that the time spent commuting (between 8 and 10%) represents the maximum exposure time for all users according to Dons et al., 2011. In this pilot study, commuting by bike and on foot is confirmed as the moment of maximum exposure for citizens.

It is important to highlight for the representativeness of the sample, that different socio-economic characteristics can be related to different habits (e.g., tobacco smoking, diet, mobility) which in turn influence the type of exposure.

Surely, a more representative sample can help to better clarify the picture of exposure on an urban scale. This means that not only a higher number of volunteers, but also the number of monitoring days per participant is more representative of people’s behavior. Then, increasing the number of days of acquisition would allow us to collect a greater amount of information to be more representative of the mix of behaviors, activities, and micro-environments experienced by the volunteers. As a future research prospect, it could be interesting, alongside the BC other particulate indicators, to better characterize all possible emission sources in the city and better understand the limits of the use of BC.

Furthermore, the use of smarter tools, such as the activity diary on a smartphone application, could speed up and facilitate the collection of daily information by volunteers.

## 5. Conclusions

The results of LL#1 were superimposed on the information obtained in LL#2 and LL#3 with the aim of integrating as much information as possible into air-quality management plans in a city heavily oppressed by vehicular traffic. The use of Living Lab and the involvement of citizens in atmospheric monitoring has helped to better quantify people’s daily exposure to air pollution, specifically with regard to BC. By superimposing the results of the three LLs, the criticalities in the urban traffic-management plans were also highlighted, as concentrations at times greater than in non-ZTL areas were found in ZTL areas.

The European Union Air Quality Directive (AQ) (EC-2008) as part of the European Green Deal requires member states to sample, analyze, and report concentrations of fine particulate matter (PM2.5) and to report the quantity of BC components contained in the measured PM2.5. Overall, information on BC measurements is largely missing from the European AirBase database (EEA, 2013). Local monitoring systems, characterized by fixed control units, often show a lack of integration between the different networks, lack of real-time data, low spatial coverage and poor temporal resolution, and also do not reproduce real-life conditions [35,36]. In current European air-quality monitoring networks, BC is a highly excluded pollutant. BC can be a valuable additional indicator of air quality to assess the risks to human health caused by primary combustion particles, and its key role is strongly highlighted by urban laboratories that allow researchers to trace its concentration and variability in a real-life context. The potential of portable sensors, continuous measurements with high temporal resolution, and the involvement of city users allows the integration of Living Labs as a necessary tool for the improvement of air quality and traffic-management plans at the local level [37]. By involving the public, healthy volunteers and volunteers with respiratory diseases and experimenting with interventions on a local scale in European cities, it is possible to scale the impacts caused by air pollution, toward broader solutions that adopt new indicators such as BC. Through the development of the LLs in the city of Turin, it was observed that without looking at local problems, designing and validating them, it is not possible to achieve the optimal solutions for a sustainable city.

## Figures and Tables

**Figure 1 ijerph-19-03866-f001:**
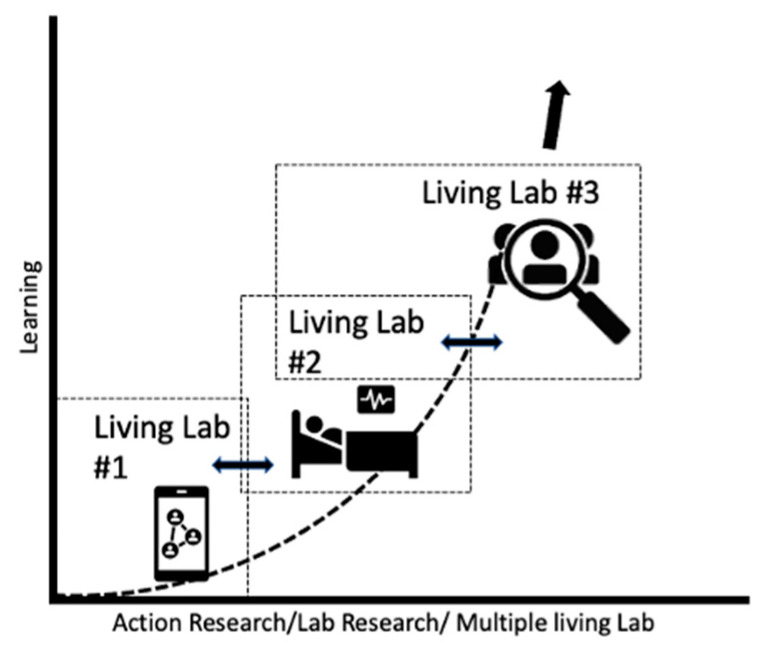
LLRA in Torino city. Multi-living lab and data interoperability: increase in knowledge.

**Figure 2 ijerph-19-03866-f002:**
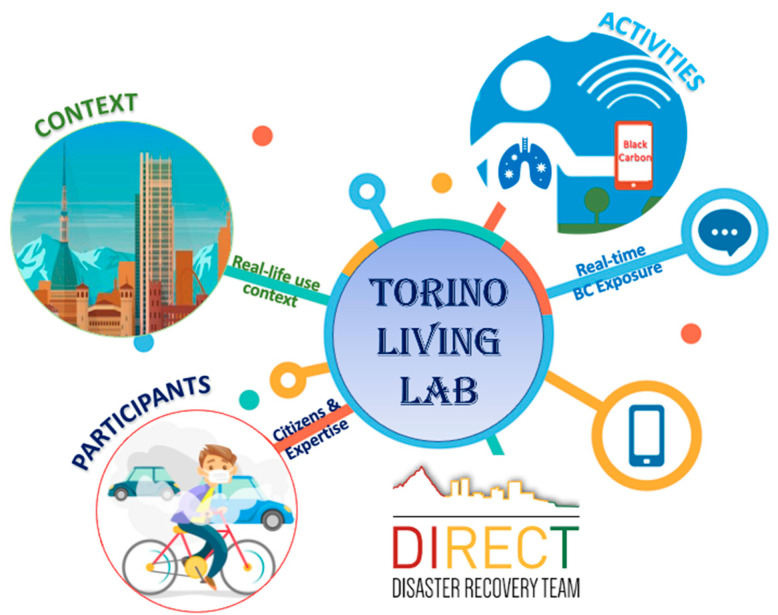
Torino Living Lab#1. LL#1 Project Logo.

**Figure 3 ijerph-19-03866-f003:**
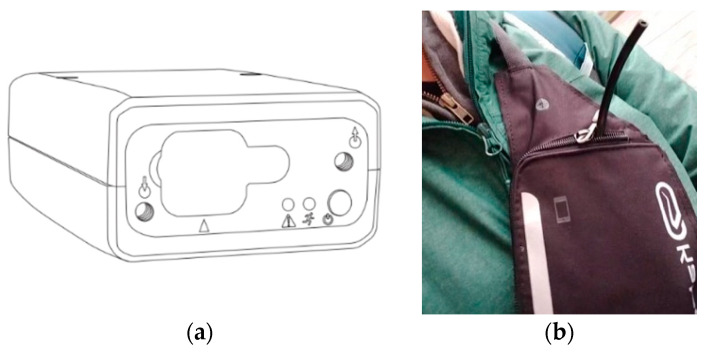
(**a**) Micro-aethalometer AE51 (Aethlabs-San Francisco, CA, USA). (**b**) Equipped shoulder bag with the sampling tube worn by volunteers.

**Figure 4 ijerph-19-03866-f004:**
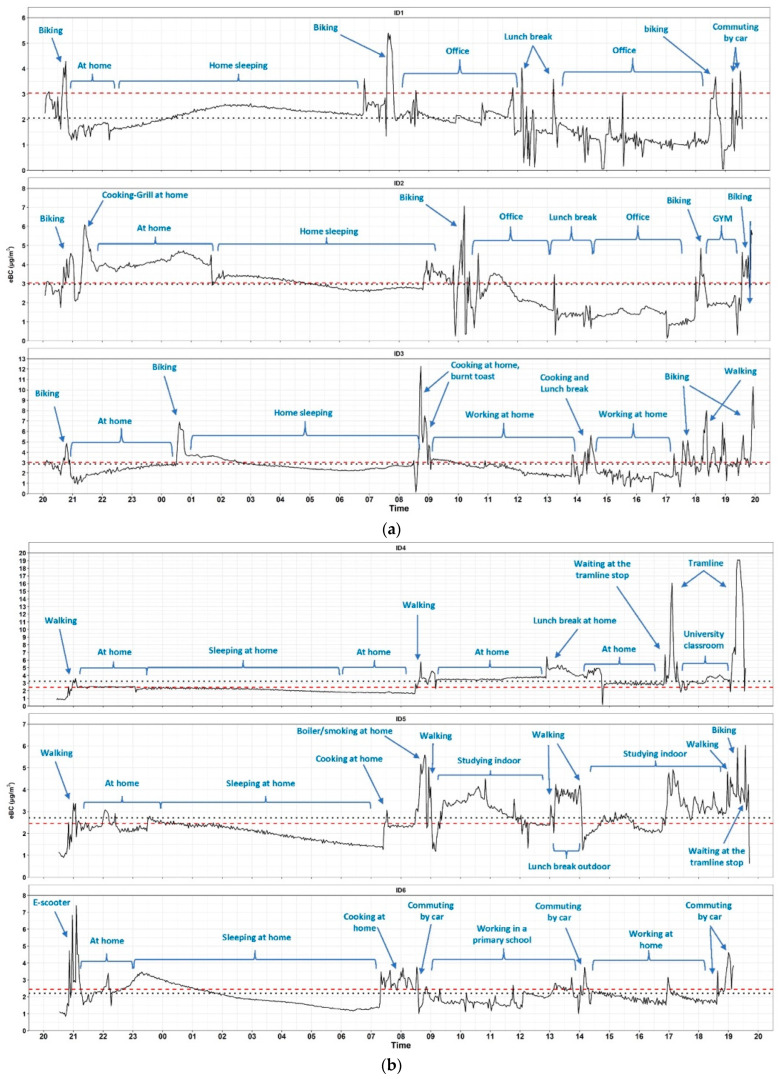
Twenty-four hour time series of personal equivalent black carbon (eBC) exposure concentrations. the sub-figure (**a**) shows the trends of the BC concentrations of ID1, ID2, ID3. The sub-figure (**b**) shows the trends of the BC concentrations of ID4, ID5, ID6. The sub-figure (**c**) shows the trends of the BC concentrations of ID6, ID7, ID8. The sub-figure (**d**) shows the trends of the BC concentrations of the ID9, ID10, ID11 and the sub-figure (**e**) the trends recorded by the ID12, ID13, ID14. (Red benchmark: average concentration measured in the fixed station of Lingotto. Black benchmark: average concentration of eBC measured by the AE51 aethalometers).

**Table 1 ijerph-19-03866-t001:** Users’ general information (ID).

Users’s General Information (ID)																
Personal	ID	01	02	03	04	05	06	07	08	09	10	11	12	13	14	15
	Gender	m	f	f	f	f	m	F	f	m	m	f	f	f	f	f
	Age	32	31	26	23	30	32	40	37	30	30	30	27	30	25	30
	Occupation	E	E	E	E	E	E	E	E	E	E	E	E	E	E	E
	Smoker													x		
Home	Home floor	6	4	1	2	2	GF	7	1	2	GF	5	1	3	5	5
	Type of road facing home	Se	ZTL	Re	Re	Re	Pr	Se	Re	Pe	Re	Se	Se	Te	Pr	Re
	Heating	IB-T	IB	IB	IB	IB	EB	EB-T	EB-T	T	EB	IB	EB	IB	EB	IB
	Other smoker at home				x	x								x		
	Use of candles or similar															x
	Distance from AQN station (km)	3.3	4.0	3.8	2.8	3.8	5.7	5.5	5.9	4.8	2.8	3.7	6.0	5.7	8.1	6.3
JanTransport	N. Trips	4	4	6	3	6	4	6	8	2	4	2	2	5	8	7
	Time (minutes)	38	70	80	70	55	85	139	260	30	50	56	25	210	196	120
	Bike	x	x	x		x					x			x		x
	Car	x					x	X	x					x	x	
	Subway								x						x	
	Foot	x	x	x	x	x	x	X	x	x	x	x	x	x	x	x
	E-Scooter						x									
	Tram				x											
Mean eBC (µg/m^3^)	At home (night) *	2.2	3.8	3.1	2.3	2.2	2.3	2.8	1.9	1.6	5.2	1.1	1.5	4.8	2.2	8.1
	Transport	3.4	3.7	3.8	7.5	3.6	3	4.4	4.6	2.8	4.1	3.6	4.6	4.8	6.9	5.2
	tot	2.1	3.0	2.9	3.2	2.7	2.2	3.7	2.8	1.6	3.9	1.6	1.6	4.7	3.3	6.7
	AQN	3.0	3.0	3.0	2.5	2.5	2.5	2.9	2.9	2.9	2.6	2.6	2.6	3.5	3.5	3.5

Legend: f (female), m (male), E (occupied), GF (ground floor), IB (internal boiler), EB (external boiler), T (district heating), AQN (Air Quality Network station). The road classification is based on open-street map (https://www.openstreetmap.org/, accessed on 1 January 2022): Pr (primary), Se (secondary) and Te (tertiary) roads are major roads. Other acronyms: Re (Residential), Pe (Pedestrian), ZTL (limited traffic zone). * We reported only night-time values because all the volunteers spent most of their time at home during this time window, whereas the conditions become too heterogeneous if considering the rest of the monitoring period.

**Table 2 ijerph-19-03866-t002:** Statistics of the measured eBC (μg/m^3^) according to the different macro- and micro-environments/activities.

MEs	µEs/Activities	Minutes	Mean	SD	Percentiles	
					1st	99th
Home	Sleeping	6774	2.8	1.8	0.6	9.4
	Other	4678	3.4	2.9	0.4	12
	Working	1628	2.5	1.1	0.6	5.5
	Cooking	200	3.6	2.4	0.5	12.8
	Tot	13,280	3.0	2.2	0.6	9.8
Other indoor	Workplace	3474	2.0	1.3	0.1	5.9
	Studying	782	3.1	0.9	1.1	5.5
	Other	824	2.2	1.5	0.3	8.1
	Bar/restaurant	297	8.3	12.2	0.4	76.6
	Physical activities	239	2.7	3.2	0.3	10.1
	Tot	5718	2.6	4.0	0.1	16.4
Transport	Walk	722	4.5	4.6	0.1	23.8
	Car	342	4.9	5.5	0.1	21.5
	Bike	299	4.3	2.9	0.3	15.2
	Subway	78	5.4	2.3	1.5	11.9
	E-scooter	25	3.7	3.3	0.9	20.9
	Tram	20	16.7	5.2	3.6	23.7
	Tot	1486	4.8	4.7	0.1	22.2
Other outdoor	Tot	194	4.0	2.9	0.7	15.1

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
