# Peer review of "Living Lab Experience in Turin: Lifestyles and Exposure to Black Carbon"

_ijerph, 2022, doi:10.3390/ijerph19073866_

Round 1

Reviewer 1 Report

The manuscript entitled "Living Lab experience in Torino city: lifestyles and exposure to black carbon" by Salimbene, et. al. reports interesting information and original findings regarding the individual monitoring of daily exposure to air pollutants. The preliminary results are promising. However, some aspects require to be improved, not only in terms of the results but also in terms of writing.

Abstract:

The aim and the scope of the work are unclear in this part. The main results in the abstract are the number of measurements and the average daily personal exposure to eBC, variables that do not reflect the most important preliminary findings of the study. If the novelty is the "Living Lab experience" concept, highlighting the exposure to lifestyle (according to the title), why some of these aspects are not described in the abstract?

Introduction: The background is not well managed. Authors incorporate information that deviates the reader's attention. There is a lot of concepts that are not relevant for the work at this point (for example, monetary quantification of health effects, Sars-CoV2 is mentioned but without context for support the Living Lab definition, Global Warming, and Climate Change, to mention some aspects). Perhaps the most important information is the description of previous studies trying to implement Living Lab studies. However, references are extensively described with a lack of connection between studies (results in concordance or opposite to preliminary results presented). This information is more suitable for the discussion section, emphasizing the evidence that supports the results obtained.

Figure 1 in the Introduction section needs to be accompanied by a figure caption explaining it. After reading the paragraph where the figure is mentioned, and seeing the figure, I didn't find the connection or the utility of the graphic.

Materials and methods section: 

The study design is not described appropriately. Why did the study involve 15 volunteers? The inclusion criteria are not described. Table 1 describes some characteristics of the volunteers, but 1) the age of volunteers has a wide range (from 23 to 40 years old) without justification, and 2) Gender is unbalanced, 73% of your population is female. Also, Table 1 is part of the results, not the Methods.

Again, Figure 2 in the Introduction section needs to be accompanied by a figure caption explaining it, not only referring to it. The Figure is too complex to be a logo, and again, it is unclear without context. Perhaps a timeline considering the time of monitoring and the activities described in all volunteers could be more useful to describe the measurements. Figure 3 could also be incorporated into the suggested timeline, to be descriptive of the methods. It took my attention why the authors didn't describe the period (Season) where measurements were performed. They describe the average temperature and some weather aspects, but as they know, certain seasons (and by extension weather conditions) are associated with high exposure to air pollutants.

Results section:

I don't see relevant differences in the results. Also, the way the results are described is confusing, and, from my point of view, anecdotic. Authors refer to eBC concentrations individually in some cases, and I perceive this is because they didn't find an important difference between individuals or eBC concentrations associated with micro-environment activities. Table 1 shows mean eBC concentrations in different activities, but I don't see the description of the differences in some of them (indoor such as bar/restaurants, and outdoor such as tramline). Where are the p values?

Figure 5 is entirely descriptive of the time series but does not show clearly the results (Tables are for that). One example is enough to visualize how the eBC exposure changes during activities/day, and the other personal time series could be in the manuscript as supplementary figures.

Discussion section:

The authors refer to the results again, which is not part of the discussion. In the beginning, they refer to some studies to contrast their results, but after a few lines, they describe all results, making some inferences that lack of support. As the authors mentioned, the study has important limitations, but I think that the most important issue to consider is the "experimental" (based on the Living Lab concept) design. The inclusion and exclusion criteria of volunteers should be discussed widely.

Final comments:

The manuscript needs to be carefully reviewed by all authors since I found multiple inconsistencies.

Some characters are in black and another font size. 

The in-text citation is mixed. Sometimes is numbered (Vancouver), sometimes is APA (author(s), and year). These inconsistencies need to be reviewed carefully.

English needs to be improved. The writing style needs to be reviewed by all authors since misspelling is constant and some phrases are redundant.

Author Response

Attached the file word. Thanks for the suggestions and corrections.

Reviewer 2 Report

The abstract requires improvement by adding more explanation regarding the purpose of the study, methods, and results. It should be more informative.

The methodology was not clearly stated. The authors have to tune it up, especially on the methodology, significance of the study, and the discussion of the results. 

Some specific comments highlighting areas of the strength and weakness of this paper as below:  

The structure of this manuscript needs to be improved. The introduction needs to be improved significantly. The introduction should focus on the review of previous work and the significance of your work. Section 1 - Introduction is a mixture of introduction and methodology and should be separated into two sections.  

I think many scholars know this living laboratory. So, why didn’t you review the previous papers on studying this living laboratories ? In a word, you need tell the readers the special points of you work.  

The strength of the work can be linked with the methodology follows the general direction. Species description, concise and straightforward. Generally, the use of subheading made readability easier.  

This an interesting paper that could make a significant contribution to the already existing science and state-of-the-art. The authors proposed an innovative approach to design a urban air quality management plan.  

The manuscript shows an organized research by the authors and the results are interesting to the researchers. These new tools would help many researchers in the world to better understand the mechanism of the design a urban air quality management plan.

Author Response

(The authors gave the same response as above.)

Reviewer 3 Report

This manuscript tests the way in which assessing exposure to air pollution by using three portable aethalometers (AE51).These are interesting ideas and there are notable strengths to the research.However,I have a number of reservations that temper my enthusiasm for the work. Regarding the introduction, I was unclear as to the rationale for focusing on Black Carbon. There are too many grammatical and typographical errors in the manuscript, the reference should keep the same. On the first page, for example. Ravina et al. should not have a comma after Ravina. In sum, while I think there is potential in this line of research, I have a number of concerns about the analyses and the validity of the tasks that temper my enthusiasm for the present work. I believe this is a good example of work that would benefit from careful adherence to best practices, and potentially more representative samples

Author Response

(The authors gave the same response as above.)
